

# Transgenerational effects on development following microplastic exposure in *Drosophila melanogaster*

Eva Jimenez-Guri[1,2], Katherine E. Roberts[1], Francisca C. García[1], Maximiliano Tourmente[3,4], Ben Longdon[1] and Brendan J. Godley[1]

[1] Centre for Ecology and Conservation, College of Life and Environmental Sciences, University of Exeter Cornwall Campus, University of Exeter, Penryn, Cornwall, United Kingdom
[2] Biology and Evolution of Marine Organisms (BEOM), Stazione Zoologica Anton Dhorn, Naples, Italy
[3] Institute for Biological and Technological Research (IIByT), National Scientific and Technical Research Council (CONICET), Córdoba, Argentina
[4] Centre for Cell and Molecular Biology. Faculty of Exact, Physical, and Natural Sciences, University of Córdoba, Córdoba, Argentina

## ABSTRACT

**Background**. Plastic pollution affects all ecosystems, and detrimental effects to animals have been reported in a growing number of studies. However, there is a paucity of evidence for effects on terrestrial animals in comparison to those in the marine realm.
**Methods**. We used the fly *Drosophila melanogaster* to study the effects that exposure to plastics may have on life history traits and immune response. We reared flies in four conditions: In media containing 1% virgin polyethylene, with no chemical additives; in media supplemented with 1% or 4% polyvinyl chloride, known to have a high content of added chemicals; and control flies in non-supplemented media. Plastic particle size ranged from 23–500 μm. We studied fly survival to viral infection, the length of the larval and pupal stage, sex ratios, fertility and the size of the resultant adult flies. We then performed crossings of F1 flies in non-supplemented media and looked at the life history traits of the F2.
**Results**. Flies treated with plastics in the food media showed changes in fertility and sex ratio, but showed no differences in developmental times, adult size or the capacity to fight infections in comparison with controls. However, the offspring of treated flies reared in non-supplemented food had shorter life cycles, and those coming from both polyvinyl chloride treatments were smaller than those offspring of controls.

# INTRODUCTION

Plastics are found ubiquitously contaminating aquatic and terrestrial systems, but we still know little of their effects on health. Microplastics (plastic particles less than five mm, either manufactured, created by weathering of larger plastic fragments or fibers from clothing items (*Andrady, 2011*; *Browne et al., 2011*)), represent the overwhelming majority of the total plastic in the sea (*Cozar et al., 2014*; *Eriksen et al., 2014*). However, microplastics are also being recovered from freshwater settings (reviewed in *Eerkes-Medrano, Thompson &*

Corresponding author
Eva Jimenez-Guri, e.jimenez-guri@exeter.ac.uk

*Aldridge, 2015*), from lake systems (*Driedger et al., 2015*) to mountain catchments (*Allen et al., 2019*), as well as in terrestrial environments (*Rillig, 2012*; *De Souza Machado et al., 2018*).

Although lagging behind the work on marine organisms (*Bergmann, Gutow & Klages, 2015*; *Martínez-Gómez et al., 2017*; *Rendell-Bhatti et al., 2020*), there is growing evidence for impacts on terrestrial animals. Several studies found birds to have ingested microplastics, either directly or via trophic transfer (*Holland, Mallory & Shutler, 2016*; *Zhao, Zhu & Li, 2016*), and earthworm growth rate was reduced and mortality increased when exposed to <150 µm polyethylene (PE) microplastic particles (*Huerta Lwanga et al., 2016*). While not affecting its life cycle, the moth *Bombyx mori* shows behavioural alterations when treated with polystyrene (PS) nanoplastics (*Parenti et al., 2020*). Aquatic mosquito larvae have been found to be able to eat 2 µm PS beads and transfer them to the adult terrestrial life stage (*Al-Jaibachi, Cuthbert & Callaghan, 2018*). Other diptera have been exposed to dietary plastics. The life traits of the midge *Chironomus riparius* were affected by 20–100 µm mixed-polymer plastic microparticles, in particular larval body mass increased and there were changes in the shapes of larval mouth parts and female wings in treated animals (*Stanković et al., 2020*). In the same organism, on the contrary, another study found that 32–500 µm PE particle exposure reduced larval size, as well as delaying development (*Silva et al., 2019*). In the soil arthropod *Folsomia candida*, ingestion of 1–63 µm PE microplastic beads is linked to behavioural changes, with reduced velocity and distance of movement (*Kim & An, 2020*). Movement was also impaired by 0.1–1 µm PS microbead ingestion in *Drosophila melanogaster* (*Zhang et al., 2020*).

Despite this growing body of literature demonstrating negative effects, the specific effects of plastic exposure on health outcomes remain almost completely unknown (*Campanale et al., 2020*; *Hirt & Body-Malapel, 2020*). Exposure to different types and sizes of plastics has been known to modulate the immune system as shown by altered gene expression (producing stress and immune related proteins) in aquatic animals such as molluscs (*Mytilus galloprovincialis* exposed to either 3 µm PS or 1–50 µm PE (*Capolupo et al., 2018*; *Détrée & Gallardo-Escárate, 2018*) and fish (*Danio rerio* exposed to 700 nm PS beads (*Veneman et al., 2017*) and *Sparus aurata* exposed to 40–150 µm polyvinyl chloride (PVC) (*Espinosa, Cuesta & Esteban, 2017*)). 130 µm PVC particles increased phagocytic activity of immune cells in the annelid *Arenicola marina* (*Wright et al., 2013*)), and PS nano and microparticles increased immune cell counts the crustacean *Daphnia magna* (*Sadler, Brunner & Plaistow, 2019*). Moreover, immune inflammatory response was triggered in molluscs (80 µm PE particles) and fish (0.5 and 50 µm PS beads) (*Von Moos, Burkhardt-Holm & Koehler, 2012*; *Jin et al., 2018*). Other immune reactions have been detected after 250–1,000 µm PE particle ingestion in earthworms (*Eisenia Andrei*) (*Rodriguez-Seijo et al., 2017*) and in human cell lines exposed to 0.5–100 µm PS beads (*Hwang et al., 2020*).

How these physiological responses translate to the immune capacity of the animals is not known, but they have been regarded as potential interference with resistance to disease (*Greven et al., 2016*). The objective of this study was to test the effect of microplastic contamination in the resistance to viral infection using the amenable model *D. melanogaster*, as well as observing their effect in the life cycle of the flies which was not previously known.

## MATERIALS AND METHODS

### Plastic particles

Polyethylene (PE) particles were obtained by grinding commercial PE nurdles (Merck, catalogue number 428043). Polyvinyl Chloride (PVC) particles were obtained by grinding commercial white PVC prime virgin plasticised pellets kindly provided by Northern Polymers and Plastics Ltd. (UK). PE nurdles are virgin (additive-free) particles, while PVC are plasticised pellets with potentially high levels of hazardous chemical additives such as phthalates (*Navarro, Gómez Tardajos & Reinecke, 2010*) and polyaromatic hydrocarbons (*Rendell-Bhatti et al., 2020*). Both pellets were ground using a SPEX Sample Prep Freezer/Mill 6870 cryogrinder using 4 cycles of two minute runs at 15 beats per second with a cooling time of 2 min between cycles. Both types of nurdles had similar mechanical properties in grinding. Fragments were sieved through 500, 250, 125 and 23 µm meshes, and a mixture was assembled (10% of 250–500 µm particles, 80% of 125–250 µm and 10% 23–125 µm). Microplastic particles were added to *Drosophila melanogaster* cornmeal media at either 1% PE, 1% PVC or 4% PVC (weight/volume, w/v) in the F1 only. Controls were reared in non-supplemented cornmeal media, as were all flies in the F2. Cornmeal media recipe and nutritional information is provided in the supplementary materials (Table S1).

### F1 matings

*D. melanogaster* mating cages were set up with apple juice 3% agar plates where females laid enough eggs to obtain the desired large amount of synchronous embryos. All embryos were pooled together and suspended in water. 10 µl embryo squirts were added to each vial, alternating conditions so one vial of each condition was prepared each time, for a total of 30 vials per treatment. Embryo squirts allow a consistently similar number of embryos to be added in each vial. Flies were left to develop at 25 °C, 12 h:12 h day night cycle, 70% humidity.

### F1 life history measurements

30 tubes for each control, 1%PVC and 1%PE treatments were used to study pupation and eclosion times. For each tube, time of pupation of every larva and time of eclosion of every adult was recorded. Offspring of every treatment were pooled to determine the ratio of males and females. For experimental set-up reasons, as we could not obtain 30 tubes for this condition, 4% PVC was not used to study developmental times or sex ratio. 4% PVC was however used for viral experiments and size measurements, as well as to set up matings to study fertility and F2 life history.

### F2 matings

To be able to assess fertility of the exposed flies, 3-day old female and male flies for each treatment obtained from the life cycle experiment were crossed at a ratio one female to two males per vial, in 20 vials. Food supplied was cornmeal media with no addition of plastic particles, and rearing conditions were 25 °C, 12 h:12 h day night cycle, 70% humidity. After 30 h, males were removed. Females were changed to a new vial every three days, up to day 15, when the female was discarded from the last tube. This way, five tubes with offspring were obtained for each female, unless she died before the end of this period.
### F2 life history measurements

For each vial, we recorded time of first pupation and of first adult eclosion. Each tube was kept for 8 days after the first adult emerged to allow eclosion of all pupae. At day 8, all adult flies from that tube were counted, discerning between males and females. Adult size, total number of flies obtained per female, as well as ratio of male and female, and total number of flies obtained per treatment were investigated.

### Size

Adult size was approximated by wing length (*Sharma, Tregenza & Hosken, 2011*) for F1 and F2 flies. Adults were kept in ethanol until both wings of 30 males for each treatment were dissected, flat mounted on a slide and measured under a Leica DM IL LED microscope using the LAS X software to undertake measurements.

### Infection survival

Virus infection survival experiments were performed on 60 F1 males of the same age for each treatment. Due to logistical constraints, only the F1 was subject to infection. Viral treatments included non-injected males (control), control injected males (sterile challenge with Ringer's solution) and virus injected males (viral challenge) for each media treatment (control, PE 1%, PVC 1% and PVC 4%). Males were kept in groups of ten. Two-day old flies were anesthetized on $CO_2$ and inoculated using a 0.0125 mm diameter stainless steel needle that was bent to a right angle *ca.* 0.25 mm from the end (Fine Science Tools, CA, USA). The bent tip of the needle was dipped into the *Drosophila* C virus (DCV) strain B6A (TCID50 $= 6.32 \times 10^9$) which is derived from an isolate collected from *D. melanogaster* in Charolles, France (*Jousset et al., 1972*). At the dose used, 100% of the flies are successfully infected (*Longdon et al., 2013*). Flies were inoculated through the pleural suture on the thorax (*Longdon et al., 2018*). We selected this route of infection as oral inoculation has been shown to lead to stochastic infection outcomes in *D. melanogaster.* However, once the virus passes through the gut barrier, both oral and pin-pricked infections follow a similar course, with both resulting in the same tissues becoming infected with DCV (*Cherry & Perrimon, 2004*; *Arnold, Johnson & White, 2013*; *Ferreira et al., 2014*; *Chtarbanova et al., 2014*). Infection of DCV by inoculation is highly pathogenic in adult *D. melanogaster*, causing increased mortality, metabolic and behavioural changes and nutritional stress in the intestine, producing similar pathologies to starvation (*Christian, 1987*; *Arnold, Johnson & White, 2013*; *Chtarbanova et al., 2014*; *Vale & Jardine, 2017*). Survival was recorded daily until all infected animals had died.

### Statistical analysis

To determine whether the different media treatments caused a significant effect on adult size and larval and pupal stages duration we applied a Kruskal-Wallis analysis of the variance because data normality and homoscedasticity were not met. The multiple comparison Dunn's test was employed to determine the cases with significant differences ($P < 0.05$). All tests were performed using the R software, version 3.5.1 (*R Core Team, 2019*).

To assess whether plastic exposure affected female fertility in the F1, we applied a generalized linear model with zero-inflated negative binomial distribution (log link

function), using total adult offspring as response variable and plastic treatment as fixed effect. We then used a Tukey test to perform multiple pairwise comparisons between treatments.

To evaluate how a rearing environment containing plastics influenced the ability of the resulting adults to resist an immune challenge with a viral pathogen (DCV), we used a parametric survival analysis (the *survreg* function in R (*Therneau, 1999*)). The maximal model included, the response variable of the number of days that each individual survived after the viral challenge, control sham infection or non-challenged was administered, any individuals who survived beyond 22 days were censored and recorded as such. The fixed effects included the plastic treatment, the viral treatment and the replicate. The Weibull distribution was used for analysis as it showed the best fit with minimum error deviance. There was no effect of replicate in the maximal model ($\chi^2 = 1361.14$, $df = 1361$, $p = 0.94$), and was removed from further analysis.

## RESULTS

### Virus resistance

In order to determine if plastic elicits an effect on the ability of flies to resist viral infection, we performed a viral resistance assay across individuals that had been reared on each of the plastic treatments. The survival of virally challenged adult flies was compared to those that were sham challenged with a virus free inoculation ('sterile control') and those that were completely naïve ('unchallenged control'). After two weeks, most virus-injected flies had died across all treatments, while most non-injected and blank-injected flies were alive (Fig. 1). To observe if there was an effect of plastic treatment we looked at the interaction between the viral treatment and plastic rearing environment, which appeared to be non-significant (Viral Treatment × Plastics: $\chi^2 = 10.55$, $df = 710$, $p = 0.103$).

### Effect of plastic exposure on *D. melanogaster* life cycle

We recorded the number of pupae and adults, discerning for male and female adults, for each treatment, as well as the time to pupation and the time of eclosion (in days). Having added the same number of eggs per vial in each treatment (see 'Materials and Methods'), we obtained 814 pupae and 810 adults for the control treatment. For the PVC treatment we obtained 691 pupae and 672 adults, and for the PE treatment we obtained 691 pupae and 677 adults. There is an asymmetry in the number of animals (pupae and adults) obtained from each treatment; plastic treatments yield 0.85 flies for every fly in the control treatment. However, these differences were non-significant (unpaired *t*-test between treatment pairs). The percentage of adults that emerged from the pupae was equivalent in all treatments, with success rates 99.3% in the control, 97.8% in 1% PE and 97.5% in 1% PVC treatments. The time of pupation follows a distribution with a peak of individuals pupating on day 3, with no significant differences between treatments, and the larval stage duration is similar for all treatments ($X^2 = 3.27$, $p = 0.195$, Fig. 2A). The same is seen for the time of eclosion of adults, also with a peak of individuals emerging on day three for all treatments, and pupa duration similar for all treatments ($X^2 = 1.73$ $p = 0.421$, Fig. 2B).

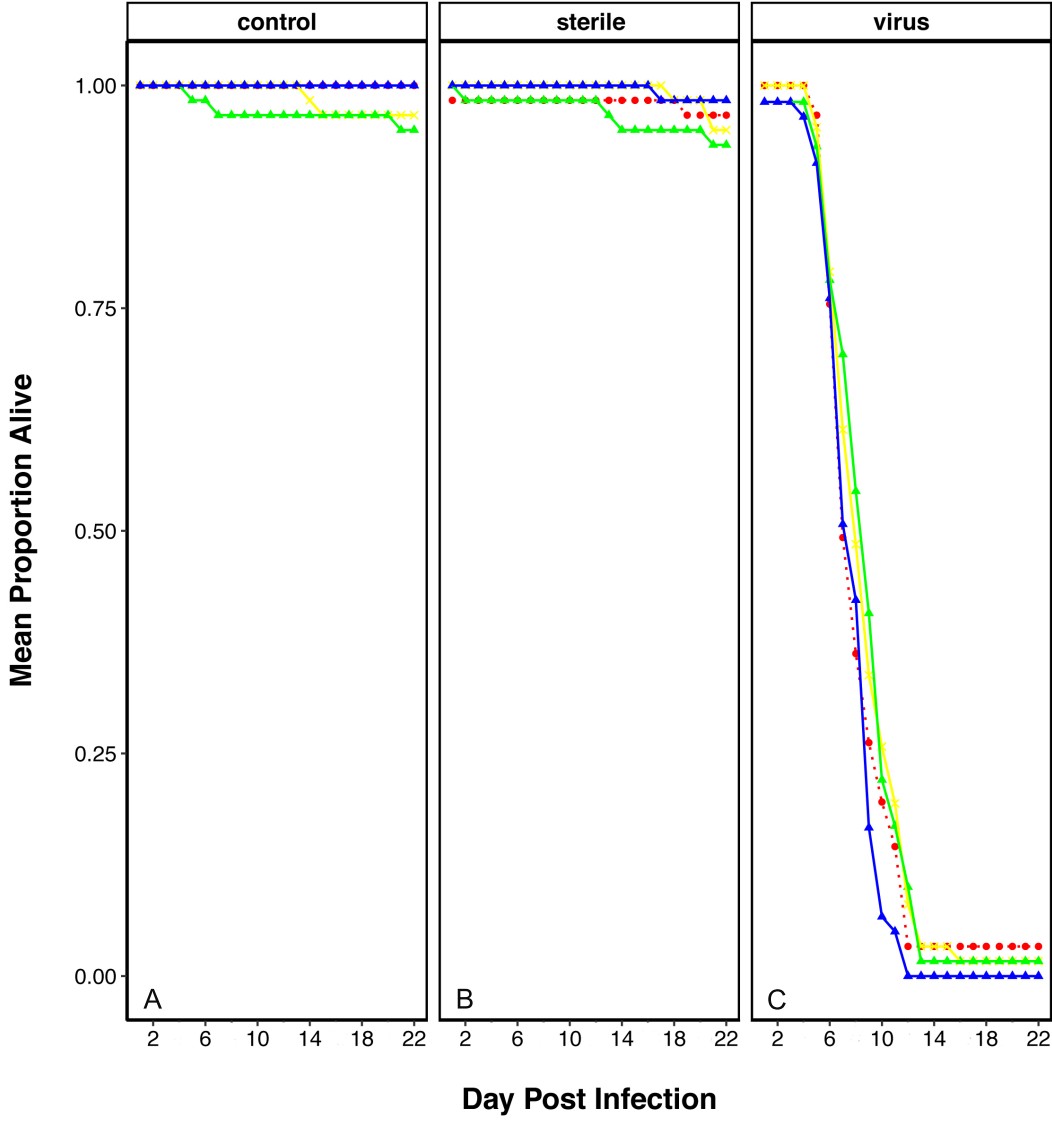

**Figure 1** **Survival of *D. melanogaster* after rearing on food containing plastics and challenged with viral pathogen (DCV).** Survival shows as mean proportion alive per day across the replicates. Red dashed line and circle: control food, yellow crosses and line: 1% virgin polyethylene (PE), green line and triangle: 1% Polyvinyl Chloride (PVC), blue line and triangle: 4% PVC. Panels are split into the three viral treatments: (A) Control- no challenge, (B) Sterile- flies challenged with a sterile infection and (C) Virus- challenged with DCV virus.

We also looked at the stage duration of larvae and pupae in the F2, where flies were reared from treated flies in non-treated food. In this case, offspring of 4% PVC treated flies (not included in the F1 life cycle experiments) had shorter larval stages ($X^2 = 43.41$, $p = 2e-09$, Fig. 2C), and offspring of all plastic treatments had shorter pupal stages than the control ($X^2 = 11.31$, $p = 0.01$, Fig. 2D).

We looked at the fertility per female in the F1 (see materials and methods). We found that females reared with 1% PE laid more eggs than the controls, and those reared with 4%
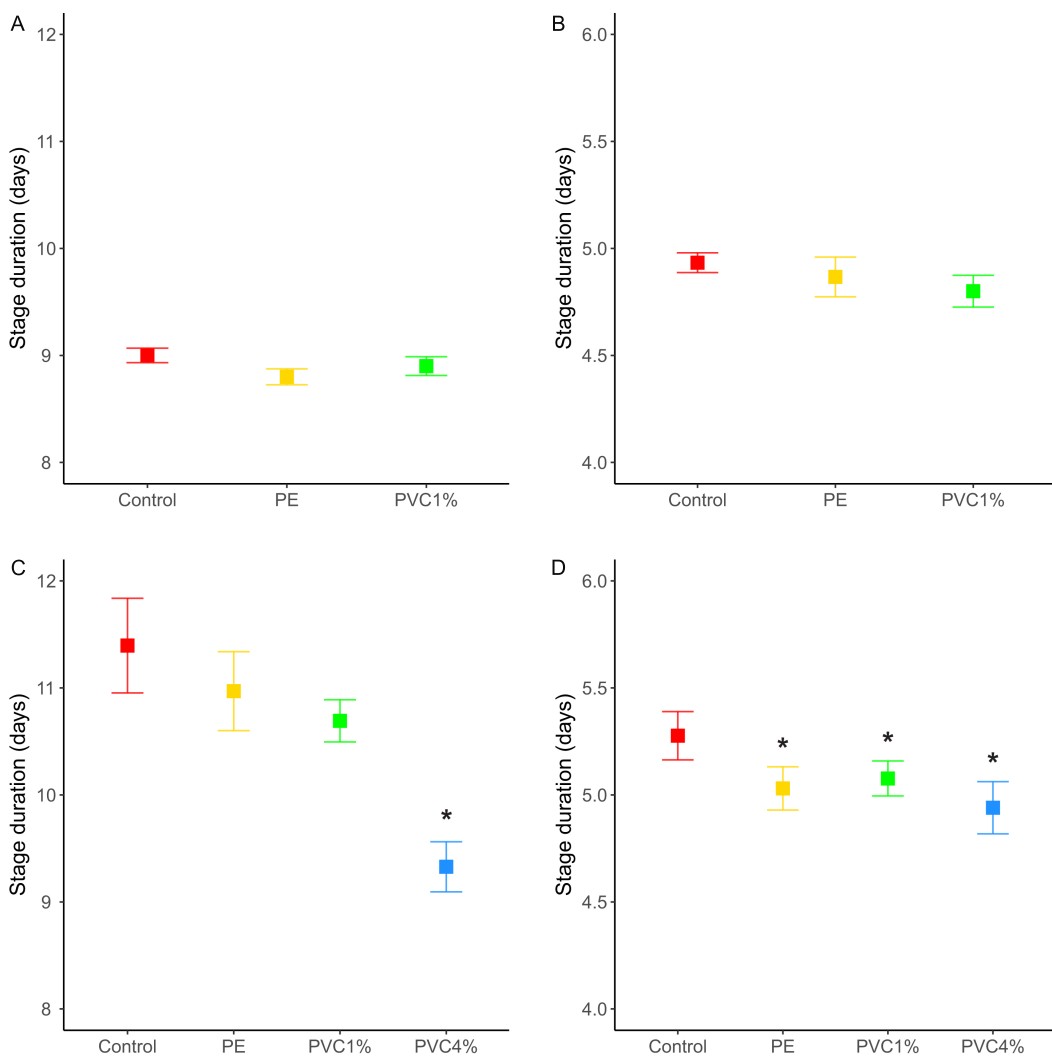

**Figure 2  Stage duration under each condition.** Stage duration in the F1 for (A) larvae and (B) pupa, and in the F2 for (C) larvae and (D) pupa for each treatment. Squares represent the mean and whiskers represent the standard error of the mean (SEM). Red: control, yellow: virgin polyethylene (PE), green: 1% polyvinyl chloride (PVC), blue: 4% PVC. Note that $y$ axis extent is different between A/C and B/D. Asterisks correspond to significant differences ($p < 0.05$) between the treatment and the control in a multiple comparisons test.

PVC laid marginally more eggs than the control (difference among all groups $X^2 = 14.4$, $p = 0.002$, Fig. 3). We saw no differences between 1% PVC treatments and controls.

The total number of females and males obtained for each treatment was recorded (Table S2 ). In the F1, while in the control treatment there was near equivalence (50.2% males), in the plastic treatments there was a slight decrease in the proportion of males. A binomial test indicated that the proportion of males of 46.4% for 1% PE and 44.5% for 1% PVC were lower than the expected 50% ($p = 0.03$ and $p = 0.002$, respectively (1-tailed)). In the F2, where larvae obtained from the treated animals were fed with food with no
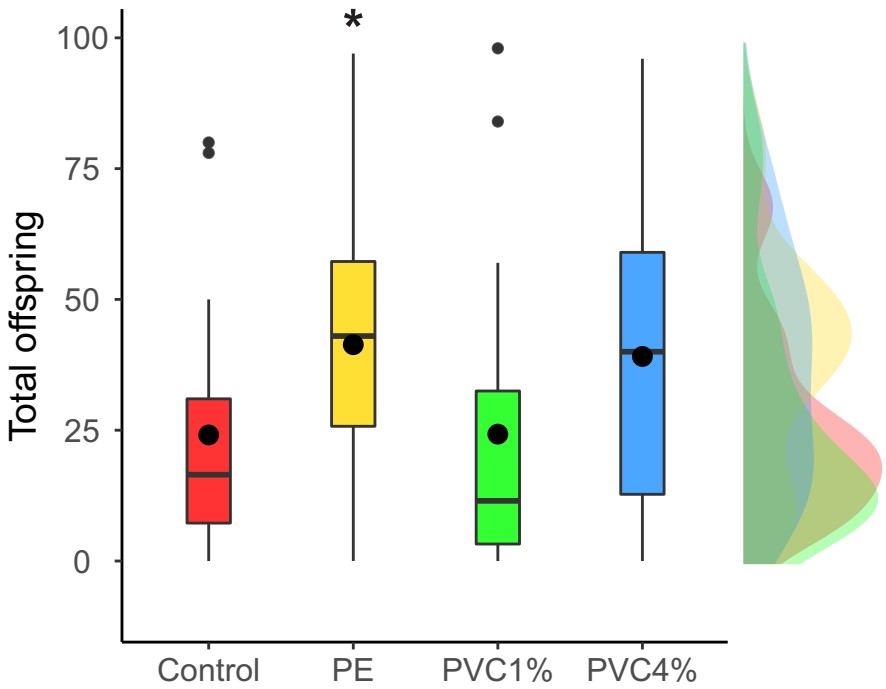

**Figure 3** **Offspring per female in the F1.** Total offspring per female, where flies reared in plastic supplemented or control media laid eggs on non-supplemented food. Boxes correspond to the interquartile range (IQR), whiskers extend to the largest value no further than 1.5 * IQR from the limit of the box, black dots represent the mean, black bars represent the median, and empty dots represent outlier values (exceeding three standard deviations from the mean). Profiles at the right of each panel illustrate the distribution density of each treatment. Red box/profile: control, yellow box/profile: 1% virgin polyethylene (PE), green box/profile: 1% polyvinyl chloride (PVC), blue box/profile: 4% PVC. Asterisks correspond to significant differences ($p < 0.05$) between the treatment and the control in a multiple comparisons test.

additives, however, we saw no differences and sex ratios were at near equivalence in all treatments.

### Adult fly size

Size of male flies resulting from larvae exposed to the different treatments was measured, as well as F2 male flies not exposed to additives in their food. F1 males from exposed larvae did not show any significant size differences between treatments and controls ($X^2 = 2.47$, $p = 0.480$ Fig. 4A). However, when the F2 individuals were measured, adults in both 1% and 4% PVC treatments were significantly smaller than the controls and 1% PE treatments ($X^2 = 14.78$, $p = 0.002$, Fig. 4B).

## DISCUSSION

*Drosophila melanogaster* has been used in many toxicological studies and recently has been utilised as a tool for the rapid assessment of microplastic mediated toxicity (*Zhang et al., 2020*). Here we used *D. melanogaster* to assess the toxicity of two types of plastic: virgin microplastic with no added chemicals (PE) and industrial plastics known to have hazardous chemical additives (PVC). Leachates of the PVC particles used in this study
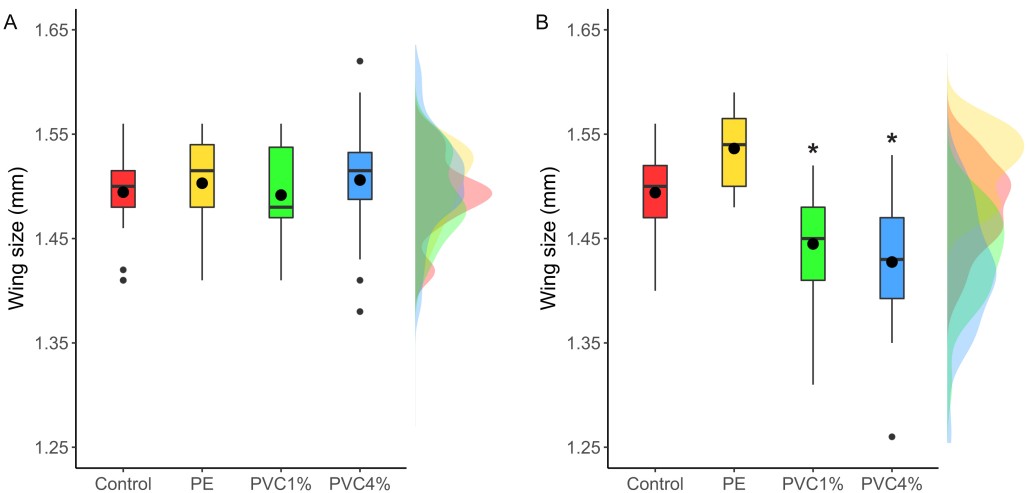

**Figure 4 Comparison of wing size under each condition.** Right wing sizes for F1 (A) and F2 (B) male flies. Boxes correspond to the interquartile range (IQR), whiskers extend to the largest value no further than 1.5 * IQR from the limit of the box, black dots represent the mean, black bars represent the median, and empty dots represent outlier values (exceeding three standard deviations from the mean). Profiles at the right of each panel illustrate the distribution density of each treatment. Red box/profile: control, yellow box/profile: virgin polyethylene (PE), green box/profile: 1% polyvinyl chloride (PVC), blue box/profile: 4% PVC. Asterisks correspond to significant differences ($p < 0.05$) between the treatment and the control in a multiple comparisons test.

have been shown to disrupt sea urchin development and to contain harmful chemicals that can be released into sea water (*Rendell-Bhatti et al., 2020*). Some plastic additives have been linked to changes in developmental time, size and fitness in *D. melanogaster* (*Quesada-Calderón et al., 2017*; *Chen et al., 2019*). In the current study, the plastics were added to a semisolid food matrix, and we did not study the possible release of chemicals from the plastics to the food. Although we were not able to test if the larvae had ingested the added plastics, the particle sizes were small enough to be eaten by later instar larvae, though they may have selectively chosen to avoid eating them. However, we saw the strongest effects in the offspring of flies treated with higher percentage of plastics and those known to have chemicals adsorbed, and we therefore hypothesise that the effects we saw may be caused by both these factors: a higher concentration of particles and higher amount of chemical additives.

It is important to note that particle size is important for the strength of any plastic effects (*Lee et al., 2013*; *Haegerbaeumer et al., 2019*; *Silva et al., 2019*). Using round 1 and 0.1 μm polystyrene beads, a study in *D. melanogaster* found gut damage and locomotion disfunction, amongst other effects (*Zhang et al., 2020*). Our study used a more environmentally relevant plastic model: plastic fragments of irregular shape and sizes ranging from 23 to 500 μm. Despite larger irregular particles being more likely to be found in the environment than nanobeads, these may be less readily available to be ingested by the larvae. The particles used in two studies on *C. riparius* had a similar size to those used in our study (*Silva et al., 2019*; *Stanković et al., 2020*). Other studies had used environmentally

relevant concentrations of plastics in riverine sediments to a concentration of 0.04% (weight in volume) (*Stanković et al., 2020*), 25 times lower than the concentration used in our 1% treatments. However, our study provides an account of what can happen in the case of food being taken from highly contaminated sites, which are now widespread. The concentration of plastics added to the food in this study, although high (1 or 4% weight in volume), is a plausible representation of such sites. This is the case in landfill sites and other highly polluted settings. (municipal solid waste sent to landfills in America contained 18.5% of plastics *United States Environment Protection Agency, 2020*), where a wide variety of insects live off food items mixed with plastic and other inedible materials (*Qasim et al., 2020*).

Our study showed no effect of the plastic treatments in the capacity to fight viral infections in *D. melanogaster*. Previous studies had reported effects in the immune system of several models by increase of immune cell numbers, physiological changes or protein or gene overexpression (*Canesi et al., 2015*; *Balbi et al., 2016*; *Greven et al., 2016*; *Sadler, Brunner & Plaistow, 2019*; *Hirt & Body-Malapel, 2020*). Other studies found injection of latex beads (composed of polystyrene) used to block phagocytosis (*Elrod-Erickson, Mishra & Schneider, 2000*) can increase susceptibility to some pathogens (*Costa et al., 2009*; *Lamiable et al., 2016*). To the best of our knowledge, this is the first report testing the outcome of viral infection after microplastic exposure. Despite our results showing no decrease in the capacity to resist viral infection in plastic treated flies, we could not rule out a transgenerational immune effect, or an additive effect after exposure for several generations, as it happens in other systems for exposures to toxicants such as heavy metals (blow fly, (*Pölkki, Kangassalo & Rantala, 2012*)) or bisphenol S (mouse, (*Brulport et al., 2021*). Further experiments with a fully-factorial design should be considered to shed light on this point.

We observed significant differences in the male/female ratio in the 1% PE and 1% PVC treatments in the F1. In contrast, no differences in sex ratios were found in *C. riparius* treated with PE particles of equivalent size to the ones used in this study (*Silva et al., 2019*). Since plastic additives often include endocrine disruptors, which are also known to affect *D. melanogaster* (*Quesada-Calderón et al., 2017*; *Chen et al., 2019*), we initially hypothesised that the male/female ratio could be affected by chemical additives in the PVC treatments. However, we also found an effect in the PE treatment, which had no added chemicals, and hence released chemicals may not be the only factor contributing to this effect. Nevertheless, we take this result with caution as the pooling of all vials from each treatment to classify offspring by sex may have confounded the results, as we could not control for vial effects.

We saw confounding changes in the fertility of treated flies. Both 1% PE and to a lesser extent 4% PVC treatments had more offspring than the control but the 1% PVC treatment showed no differences with the control. Similar confounding results were seen in *D. magna*, where exposure to 1-5μm spheres reduced fertility in successive generations after microplastic exposure (*Martins & Guilhermino, 2018*), but animals exposed to 0.5μm PS beads showed significant greater clutch sizes (*Sadler, Brunner & Plaistow, 2019*). These results were linked to microplastic induced physical effects and chemical toxicity for the

reduction in fertility, and upregulation of haemocytes (immune response) in the case of the increased fertility. Future work could inform whether these mechanisms are also responsible for the effects we saw in our case.

We saw no effect in developmental time or adult size in the *D. melanogaster* exposed to microplastics. In the dipteran *Chironomus riparius* exposure to mixed microplastics (polyethylene-terephthalate, polystyrene, PVC and polyamide) resulted in increased average body mass and length, as well as a significant increase of the development time from first instar to adult (*Stanković et al., 2020*). In contrast, ingestion of polyethylene by the same dipteran *C. riparius*, produced conflicting results showing reduced larval growth, but also a delay in the emergence of adults (*Silva et al., 2019*). It is unclear why all these results differ but it could be down to the aquatic medium in which *C. riparius* was exposed or due to the different properties of the microplastic or microplastic mixes used.

Despite not seeing timing or size effects in the F1 generation of *D. melanogaster*, we saw differences in the offspring of treated flies: the size of the F2 adults in PVC treatments (at both concentrations tested) were smaller than control animals, and the life cycle was shorter in the offspring of all treated flies. Similar results have been seen in the aquatic model *Daphnia magna*, where individuals exposed to microplastics had smaller offspring but no other changes in life history traits. This change in *D. magna* was linked to increased numbers of immune cells under microplastic exposure (*Sadler, Brunner & Plaistow, 2019*). In our case, the flies were not exposed to plastics, but were the offspring of exposed flies. In *D. melanogaster* and other flies, the size of pupae and adult flies depends on the size of the larvae at the moment of pupation, which is determined by the availability of food during larval development. Starvation usually correlates with a longer larval stage, to attain the minimal viable size to survive metamorphosis, and a shorter pupal stage, probably since smaller larvae means less tissue to be metamorphosed (*Alcaine-Colet, Wotton & Jimenez-Guri, 2015*). In our case, the affected flies were reared in plastic-free media, and therefore we did not expect that the food source would trigger a change in their body size and developmental time. The changes to smaller size and shorter life cycles in the F2 generation could be due to many potential factors such as an epigenetic effect in the F1 which affects the F2, or a condition effect in the mothers, such as physiological toxicity caused by physical effects (for example mechanical damage to the gut of the mother) or chemical toxicity, that affects offspring development. Plastic particles are known to cause oxidative stress (reviewed in *Pérez-Albaladejo, Solé & Porte, 2020*), neurotoxicity (reviewed in (*Prüst, Meijer & Westerink, 2020*)) and developmental toxicity in other animals (*Martínez-Gómez et al., 2017*; *Messinetti et al., 2018*; *Rendell-Bhatti et al., 2020*), as well as reduced gut function (*Wright et al., 2013*), which may have a negative knock on effect in subsequent generations. In favour of this hypothesis, we saw that the offspring of flies with the higher dose of plastics (4% PVC), which also are known to contain harmful chemicals (*Rendell-Bhatti et al., 2020*), were the ones showing both the strongest changes in their life cycles, with shorter larval and pupal stages, and the greatest reduction in body size.

## CONCLUSIONS

Exposure to microplastic particles affects sex ratios and fecundity in *D. melanogaster* within the exposed generation, but does not significantly change its life cycle nor affect the ability of male flies to fight viral infection. However, non-exposed offspring of exposed flies from higher content (4% w/v) PVC show a shorter larval stage compared to all other treatments and controls, and non-exposed offspring from exposed flies of all plastic treatments show a decrease in the duration of the pupa stage. Despite flies reared in supplemented food showing no differences in body size, non-exposed offspring of exposed flies from PVC treatments show a significant decrease in body size with respect to all other treatments. We saw the strongest effects in the non-exposed offspring of flies treated with higher percentage of plastics and those known to have chemicals adsorbed. Further investigation is needed into the effect of our increasingly plastic-contaminated terrestrial environments on life history traits of exposed animals.

## ACKNOWLEDGEMENTS

We thank Dave Hosken, MD Sharma and Keiko Oku for support with the F2 fertility experimental set up, Tamara Galloway and Adam Porter for access to and help with plastic grinding facilities, Flora Rendell-Bhatti for help with plastic grinding, Karl R Wotton for the use of microscope equipment and critically reading the manuscript, Marta Morey for help with fly morphology, and Pablo Capilla-Lasheras for help with an earlier version of the work. The manuscript benefitted from the constructive review of three anonymous reviewers and the Editor.

### Funding

Eva Jimenez-Guri was supported by the European Union's Horizon 2020 research and innovation programme under the Marie Skłodowska-Curie grant agreement no. 882904. Katherine E Roberts and Ben Longdon are funded by a Sir Henry Dale Fellowship jointly funded by the Wellcome Trust and the Royal Society (grant no. 109356/Z15/Z). The funders had no role in study design, data collection and analysis, decision to publish, or preparation of the manuscript.

### Grant Disclosures

The following grant information was disclosed by the authors:
Marie Skłodowska-Curie: 882904.
Wellcome Trust and the Royal Society: 109356/Z15/Z.

### Competing Interests

The authors declare there are no competing interests.
## Author Contributions

- Eva Jimenez-Guri conceived and designed the experiments, performed the experiments, analyzed the data, prepared figures and/or tables, authored or reviewed drafts of the paper, coordinated the work, and approved the final draft.
- Katherine E. Roberts conceived and designed the experiments, performed the experiments, analyzed the data, prepared figures and/or tables, and approved the final draft.
- Francisca C. García and Maximiliano Tourmente analyzed the data, prepared figures and/or tables, and approved the final draft.
- Ben Longdon and Brendan J. Godley conceived and designed the experiments, authored or reviewed drafts of the paper, and approved the final draft.

## Data Availability

The code and raw data are available in the Supplemental Files.

## Supplemental Information

Supplemental information for this article can be found online at http://dx.doi.org/10.7717/peerj.11369#supplemental-information.

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
