# Peer review of "Transgenerational effects on development following microplastic exposure in Drosophila melanogaster"

_PeerJ, doi:10.7717/peerj.11369_

## Round 0.1 · original submission · Major Revisions

Additional details of the experimental methods need to be provided in the revised manuscript. For example, it is not clear what sizes and concentrations of microplastics have been processed in the fruit fly. Moreover, the reason why only F1 males were exposed in the virus
infection survival experiment is not mentioned. Without a fully-factorial design, it doesn't appear to be possible to tease apart time-dependent effects from true transgenerational effects. The discussion section needs improvement in the interpretation of the experimental results, especially in light of the statistical design issue indicated above. Additional comments are provided by the reviewers, so there is no reason to repeat them here.

Reviewer 1 ·

Basic reporting

no comment

Experimental design

Whilst the research questions is well defined, relevant and meaningful, there are some gaps in the methods described that would make the work difficult to replicate.
These are listed below:
Line 88 - please state whether the two types of nurdles had similar mechanical properties in grinding, i.e. were they equally blunt or sharp, long etc. Where is the proof that the flies actually ate the plastic?
90-93 - can the flies avoid eating the plastic in the media? Did they eat less media altogether because they didnt like the plastic? There is evidence that animals are reluctant to eat plastic and studies on Daphnia have shown that the impact of plastic ingestion is not so much the plastic itself but the reduced ingestion of food.
94 - I am not a Drosophila breeder and did not understand where the embryos came from. Do the authors mean eggs? It is not stated how much water was added to the embryos, therefore removing a set volume is meaningless. Overall I was unsure of numbers of embryos and whether the numbers varied considerably between tubes.

Validity of the findings

This is an interesting study and gives detail of negative results as well as effects of plastics. The data are controlled, robust and sound.
Conclusions are well stated and linked to the original research question. However I remain unconvinced that the authors have ruled out reduced feeding as the source of smaller body sizes. This should be addressed and discussed.

Additional comments

Have you considered whether the fragments are mechanically damaging the gut if they are sharp?
For me the lack of information on the plastic raises some questions about what the flies were ingesting. Do you have any data that proves that they ingested the plastic?

Reviewer 2 ·

Basic reporting

no comment

Experimental design

Methods described with sufficient detail & information to replicate.
Methods should be described with sufficient information to be reproducible by another investigator.

Validity of the findings

no comment

Additional comments

no comment

Annotated reviews are not available for download in order to protect the identity of reviewers who chose to remain anonymous.

Reviewer 3 ·

Basic reporting

- The language is appropriate, with a few minor grammar issues (e.g. L20, L144, L201, L206 L270, L313) which should be resolved but do not prohibit understanding of the text in general.
- The introduction is well structured and covers the relevant literature, but the paragraph on immune responses to microplastics (L60-71) would benefit from summarizing the studies a little, e.g. by types and sizes of plastics that were used, types of responses found, directly contradictory results (same species and plastic type tested in different studies with different outcomes) versus differences between species and plastic types etc. Identifying such patterns would be much more informative for the reader than just a list of studies. Of course this article is not a literature review, however a little guidance for the reader in the introduction section would be useful.
- The article adheres to the standard section instructions by PeerJ and the information is appropriately allocated to the different sections.
- The Figures are well presented and the legends contain all necessary information to understand them.
- The raw data is supplied in an excel file, but it does not contain explanatory metadata. While the variables used are mostly self-explanatory, the data structure as such could be explained to ensure data could be reused appropriately ( e.g. explaining what empty cells versus 0 versus n/a signify, whether rows are individuals/days/…)

Experimental design

- This article clearly falls within the scope of PeerJ as it presents primary research in the biological sciences.
- The research questions are well defined and the knowledge gap of how microplastics affect infection outcomes is clearly presented and addressed by the data. To avoid confusing readers however, I recommend revising the text to avoid suggesting that immune responses to microplastics were tested since they were not explicitly looked at in this study (e.g. L163, L220). While I agree that any effects on infection outcomes would likely be mediated by immune response differences, there are other possibilities, such as altered resource availability and a reduced resource availability for both host and parasite. In return, the finding that infection outcomes are not altered by microplastic exposure does not prove that there are no effects on immune responses, they may just be inconsequential for this particular infection.
- The experimental design is rigorous and conforms to a high technical standard. The methods describe the experiments well and are sufficient except for a couple of details that should be clarified:
L89 The minimum and maximum sizes of the particles are useful, however a distribution of sizes ( at least mean +- std error) would potentially be even more useful to compare the outcomes to other studies and size ranges used therein. As the authors point out in the discussion themselves, microplastic particle size can strongly influence the effects on organisms.
L129 Does this viral injection treatment result in successful infection 100% of the time? If not, it may be more appropriate to talk about virus exposed rather than infected flies in the results. Either way, it might be useful to tell readers that do not work with this virus a few more details about the infection and its effects on the host.

Validity of the findings

- All underlying data and statistical code have been provided.
- Speculation is generally clearly marked, except for the assumption that immune responses were tested (see earlier comment in experimental design section)
- The conclusions are well founded on the research questions, results and discussion. However, they could be easier to grasp for readers if they were summarised more in the categories “within exposed generation” and “transgenerational effects in non-exposed offspring of exposed flies”. Furthermore, given the prominent reference to the result in the 4%PVC treatment, it would be good to remind the reader that this treatment was not tested in the F1 so it is at least not clear whether this is only a transgenerational effect.

Additional comments

L27-28 This sentence can be misunderstood to say that the F2 were descendants of the survivors of the infection experiment, please revise to clarify that the F2 assays were carried out in an unbiased way.
L79 The direction of the fecundity effect is not clear from the text.
L183 Presumably should say “no significant differences between treatments”

---

## Round 0.2 · accepted · Accept

Thanks for revising your manuscript based on reviewer comments. There are a couple more corrections that are needed in the text that was added.
L71 The Aljaibachi et al 2020 study did not look at immune responses – did you mean to add a different citation here?
L80 It still says here that the immune response was tested – this needs changing as in the rest of the text to say resistance to disease was tested.
L213 In the F2, where…
L287 greater

Reviewer 3 ·

Basic reporting

The authors have substantially revised and improved their manuscript. I am satisfied with those changes and only want to point out a few small mistakes in the newly added text:

L71 The Aljaibachi et al 2020 study did not look at immune responses – did you mean to add a different citation here?
L80 It still says here that the immune response was tested – this needs changing as in the rest of the text to say resistance to disease was tested.
L213 In the F2, where…
L287 greater

Experimental design

satisfied with the revision

Validity of the findings

satisfied with the revision